# Adaptor protein XB130 regulates the aggressiveness of cholangiocarcinoma

**Pirawan Poosekeaw[1], Chawalit Pairojkul[1], Banchob Sripa[1], Prakasit Sa Ngiamwibool[1], Sitthichai Iamsaard[2], Chadamas Sakonsinsiri[3,4], Raynoo Thanan[3,4], Piti Ungarreevittaya[1]***

1 Department of Pathology, Faculty of Medicine, Khon Kaen University, Khon Kaen, Thailand, 2 Department of Anatomy, Faculty of Medicine, Khon Kaen University, Khon Kaen, Thailand, 3 Department of Biochemistry, Faculty of Medicine, Khon Kaen University, Khon Kaen, Thailand, 4 Cholangiocarcinoma Research Institute, Khon Kaen University, Khon Kaen, Thailand

* pitiun@kku.ac.th

**Data Availability Statement:** All relevant data are within the paper and its Supporting information files.

## Abstract

Cholangiocarcinoma (CCA) is a group of heterogenous malignancies arising from bile duct epithelium with distinct pathological features. Adaptor proteins have implicated in cell proliferation, migration, and invasion of different cancer cells. The objective of this study was to assess whether the adaptor protein XB130 (AFAP1L2) is a critical biological determinant of CCA outcome. XB130 expression levels were investigated in four CCA cell lines compared to an immortalized cholangiocyte cell line by Western blotting. Small interfering (si) RNA-mediated *XB130* gene silencing was conducted to evaluate the effects of reduced XB130 expression on cell proliferation, migration, and invasion by MTT, transwell migration and cell invasion assay. The immunohistochemical quantification of XB130 levels were performed in surgically resected formalin-fixed, paraffin-embedded specimens obtained from 151 CCA patients. The relationship between XB130 expression and the clinicopathological parameters of CCA patients were analyzed. Our results showed that XB130 was highly expressed in KKU-213A cell line. Knockdown of XB130 using siRNA significantly decreased the proliferation, migration, and invasion properties of KKU-213A cells through the inhibition of PI3K/Akt pathway, suggesting that XB130 plays an important role in CCA progression. Moreover, elevated XB130 expression levels were positive relationship with lymphovascular space invasion (LVSI), intrahepatic type of CCA, high TNM staging (stage III, IV), high T classification (T3, T4), and lymph node metastasis. We provide the first evidence that the overexpression of XB130 is associated with tumorigenic properties of CCA cells, leading to CCA progression with aggressive clinical outcomes.

## Introduction

Cholangiocarcinoma (CCA) is a malignancy of biliary epithelial cells and its incidence has increased worldwide [1]. The risk factors of CCA are heterogeneous and differ globally [1]. Primary sclerosing cholangitis is the most typical risk factor for CCA in the Western countries,

**Funding:** This study was financially supported by Invitation Research Fund from Faculty of Medicine, Khon Kaen University, Thailand to P.P. and P.U. (IN63122) and Thailand Research Fund to R.T. (RSA6280005).

**Competing interests:** The authors have declared that no competing interests exist.

while in several Asian countries *Opisthorchis viverrini* is determined as the key risk factor for CCA [2]. CCA is classified into intrahepatic and extrahepatic types, based on its anatomical location along the biliary tree. Intrahepatic CCA develops within the small and second order large bile duct branches, which are situated in the liver parenchyma. Extrahepatic CCA involves the extrahepatic large bile duct location within biliary tree and the hepatoduodenal ligament [3]. CCA is histologically graded into well, moderately, and poorly differentiated adenocarcinomas [4]. CCA remains a major health problem which leads to high mortality in Thailand, notably in the Northeastern region [5]. Because of a high degree of malignancy, rapid progression, low surgical resection rates and high recurrence rates, CCA prognoses are poor.

Several independent prognostic factors for CCA have been identified, including serum carcinoembryonic antigen (CEA) levels, alkaline phosphatase levels, relapse of disease, intraoperative blood transfusion, lymph node metastasis, TNM staging [6, 7], histological differentiation of tumors, surgical margins, types of CCA, Caudal Type Homeobox 2 (CDX2), and human epidermal growth factor receptor 2 (*HER2)* gene amplification [8]. It has been shown that serum CEA levels of greater than 2.5 ng/mL are considered as an independent poor prognostic factor for resectable CCA patients [6, 7]. CCA patients with CDX2-positive tumors were reported to have significantly better survival than those with CDX2-negative tumors [9]. Moreover, the overexpression of *HER-2* gene amplification using chromogenic *in situ* hybridization (CISH) is an independent prognostic factor for survival in the subgroup of extrahepatic CCA patients with lymph node metastases [7]. However, the estimated overall 5-year survival rate of CCA is less than 10% and only approximately one third of the CCA patients are possible for receiving curative treatment at the diagnostic time [10]. Therefore, the search for novel prognostic marker is still needed for improvement of CCA therapy.

In the development of the human embryo, the liver, extrahepatic biliary ducts, gallbladder, and pancreas duct arise from the same epithelial anlage [11]. Accumulating evidence suggests that CCA and pancreatic ductal adenocarcinoma have similar pathology, background, and development [12]. Zhang et al. demonstrated that the high expression of XB130 is an independent prognostic marker to predict poor outcome after surgical resection of pancreatic ductal adenocarcinoma [13]. However, the roles and expression levels of XB130 protein in CCA, which is a malignancy developing from the biliary epithelial cells, have never been investigated.

XB130, which is also known as actin filament-associated protein 1 like 2 (AFAP1L2), is an important member of the actin filament-associated protein (AFAP) family of adaptor proteins. *XB130* gene is located on human chromosome 10q25.3 and encodes a protein of 818 amino acids [14]. XB130 is an adaptor protein involved in cell proliferation, survival, and migration [15–17]. This adaptor protein has been shown to cause the activation of kinases and related downstream proteins in many signaling pathways [18]. It can also promote interactions between protein binding partners, triggering signaling cascades. For instance, XB130 binds to the p85α subunit of phosphatidylinositol-3-kinase (PI3K) and activates protein kinase B (Akt). The excessive activation of PI3K/Akt signaling pathway promotes cell proliferation by regulating cell cycle during G1 to S phase progression and by inducing cell migration and tumor cell survival [19]. XB130 is mainly distributed in the cytosol, where the chemical reactions predominantly occur and is associated with physiological processes and oncogenesis of certain malignancies [18]. Functional roles of XB130 have been examined in various types of cancer both *in vitro* and *in vivo*. For example, the overexpression of XB130 in human esophageal squamous cell carcinoma (ESCC) has shown to be associated with cell cycle progression and poor prognosis [20]. The previous studies in lung and thyroid cancer cells demonstrated that XB130 is a potential regulator of tyrosine kinase-mediated signaling and substrate that control of cell proliferation and apoptosis [19, 21]. Knockdown of XB130 in human lung and thyroid

cancer cells by small interfering RNA (siRNA) suppressed cell cycle progression and induced spontaneous apoptosis [19, 21]. However, little is known about the roles of XB130 in CCA.

In this study, we aimed to investigate: i) the expression levels of XB130 in four different CCA cell lines (KKU-100, KKU-213A, KKU-213C, and KKU-023) compared to an immortalized cholangiocyte cell line (MMNK1); ii) the effects of decreased XB130 expression on cell proliferation, cell migration and invasion in XB130-expressing KKU-213A cells; iii) the expression levels of XB130 in CCA tissues using immunohistochemistry; and iv) the relationship between the XB130 expression levels with the clinicopathological features and survival rates of CCA patients.

## Materials and methods

### Cell culture

For different CCA cell lines (KKU-100, KKU-213A, KKU-213C, and KKU-023) and an immortal cholangiocyte cell line (MMNK1) were obtained from the Cholangiocarcinoma Research Institute, Khon Kaen University, Thailand. KKU-213A and KKU-213C CCA cell lines were established and characterized by Sripa B [22]. All cell lines were cultured in Ham′s F-12 (Life technologies, Grand Island, NY, USA) containing 10% heat-inactivated fetal bovine serum, 100 U/mL penicillin and 100 ug/mL streptomycin (Life technologies, Grand Island, NY, USA). Cells were incubated at 37°C in a humidified incubator with an atmosphere of 5% $CO_2$.

### Screening of the XB130 expression in cell lines using Western blot analysis

Cells were lysed on ice in RIPA buffer (150 mM NaCl, 50 mM Tris-HCl, 1% (v/v) Triton X-100, 1% (w/v) sodium deoxycholate, 0.1% (w/v) sodium dodecyl sulfate (SDS) supplemented with protease inhibitor cocktail (Thermo Fisher Scientific, MA, USA; S1 Appendix) The protein concentration was measured using Pierce BCA Protein assay kit (Pierce Biotechnology, Rockford, USA). Proteins in the total cell lysate were separated onto SDS-PAGE (4% stacking gel and 8% separating gel) and transferred onto a polyvinylidene difluoride (PVDF) membrane (Bio-Rad, CA, USA). After blocking the blot in a solution of 5% skimmed milk in Tris-buffered saline (TBS) pH 8.0 at room temperature for 1 h, membrane-bound proteins were probed with mouse anti-XB130 monoclonal antibody (1:1000; Santa Cruz, USA), rabbit anti-E-cadherin antibody (1:1000, Cell Signaling Technology, Danvers, MA, USA), mouse anti-vimentin antibody (1:1000, Agilent Dako, Santa Clara, CA, US), rabbit anti-Akt antibody (1:1000, Cell Signaling Technology, Danvers, MA, USA), rabbit anti-Ser473 phosphorylated Akt (pAkt) antibody (1:1000, Cell Signaling Technology, Danvers, MA, USA) and mouse anti-β-actin antibody (1:60000, Sigma, Louis, MO, USA) overnight at 4°C. The membrane was washed and incubated with peroxidase-labelled anti-rabbit or anti-mouse secondary antibodies (1:2000) for 1 h at room temperature. Finally, immunoreactive bands were visible to Amersham™ ECL™ Prime Western Blotting Detection Reagent (GE Healthcare, Buckinghamshire, UK) for chemiluminescence detection which was performed using an Amersham Imager™ 600 (GE Healthcare Bio-Sciences AB, Uppsala, Sweden). Original images of Western blot results are shown in S1–S7 Figs.

### Knockdown of XB130 in an CCA cell line using siRNA

Specific siRNA against XB130 (siXB130), ON-TARGET plus Human XB130 siRNAs; cat. no. L-014917-02-0005, and a negative control or scramble, ON-TARGET plus Non-targeting Pool siRNAs; cat no. D-001810-10-05, were purchased from GE Healthcare Dharmacon Inc.

(Lafayette, CO, USA). KKU-213A cells ($12 \times 10^4$ cells/well) were transfected with 50 nM of siXB130 or scramble using lipofectamine® RNAiMax (Invitrogen, Carlsbad, CA, USA) according to the manufacturer's protocol. After 48 h transfection, cells were used for detection of XB130 expression by western blot analysis as well as cell proliferation, invasion and migration assays.

## Cell proliferation assay

At 48 h after XB130 knockdown, treatment with scramble and siXB130, the cells were trypsinized and seeded onto 96-well plates ($3.0 \times 10^4$ cells) in quintuplicate. Cell proliferation was measured using the methyl thiazolyl tetrazolium (MTT) assay as previously described [23].

## Cell invasion assay

Cell invasion assay was performed using a 24-well Matrigel Invasion Chamber (Discovery Labware, Inc, Bedford, MA, USA) as previously described [24]. The cell invasion was determined at 24 h after incubation.

## Cell migration assay

The migration ability of CCA cells across the membrane filter of 8 μm pore-size was measured using a Boyden chamber in a 24-well plate (Corning, NY, USA) as previously described [23]. The number of cells migrated across the membrane was determined at 24 h.

## Human cholangiocarcinoma tissues

The formalin-fixed and paraffin-embedded CCA tissues used in this study were the leftover tissues after pathological diagnosis of CCA (n = 151 cases) in the Srinagarind Hospital, Khon Kaen University, from January 2007 to August 2017 and performed tissue microarray (4 cores, 2 mm). The protocol of this study was approved by the Ethics Committee for Human Research, Khon Kaen University (HE621409).

## Immunohistochemical staining

The immunohistochemistry (IHC) method for XB130 protein was performed on tissue microarray slide sections, using a VENTANA BenchMark XT Slide Staining System (Roche Diagnostics, NJ, USA). The tissue microarray slides were cut at 4 μm thickness and dried in a hot-air oven overnight at 50°C. Immunohistochemical staining was carried out following the manufacturer's recommended protocol. Mouse anti-AFAP1L2/XB130 monoclonal antibody (1:100; Santa Cruz, TX, USA) was used as a primary antibody and its presence was visualized using an OptiView DAB IHC detection kit (Ventana, AZ, USA). For XB130 assessment, the stained tissue microarray slides were entirely scanned to assign the scores using a ScanScope® XT digital slide scanner (Leica Biosystems, Singapore).

## Evaluation of immunohistochemical staining

The semi-quantitative analysis of IHC scores using an ImageJ software, Java-based image processing program developed at the National Institutes of Health and the Laboratory for Optical and Computational Instrumentation, University of Wisconsin [25, 26]. The quantification methods were performed following the previous report [27]. XB130 expression levels (IHC scores) of 151 CCA tissues examined ranged from 28.00 to 220.45.

## Clinicopathological parameters of CCA tissues

The clinicopathologic variables included age, gender, type of CCA, histological type, histologic differentiation, LVSI, T classification, N classification (lymph node metastasis), M classification, TNM staging, and surgical margins were used for data analysis. By anatomical location of the tumor, CCA was classified into intrahepatic CCA and extrahepatic CCA. Histologic differentiation of CCA was classified by histology into well, moderately, and poorly differentiated adenocarcinomas [4]. Histologic type was defined as papillary or tubular base on architectural growth pattern. The LVSI is the histologic visible invasion of a cancer cells into lymphatic and/or blood vessels which is the cause of lymphatic and vascular spreading. Lymph node metastasis is cancer that has spread to lymph nodes and a surgical margin is the margin of apparently non-tumorous tissue around a tumor that has been surgically removed [28]. The TNM classification of Malignant Tumors seventh edition is an anatomically based system that records the primary size, regional nodal extent of the tumor, and distance metastasis. TNM is a notation system that describes the stage of a cancer, T category describes the primary tumor site, N category describes the regional lymph node involvement and M category describes the presence or otherwise of distant metastatic spread [29].

## Statistical analyses

The relationship between XB130 expression levels with clinicopathological features were performed by linear regression. Survival analysis was performed using the Kaplan-Meier method and the difference between survival curves was calculated using the log-rank test. Cox's proportional hazards models of univariate and multivariate analyses were performed to define risks and identify independent prognostic factors of XB130 protein expression. Univariate and multivariate logistic regression analyses were performed to identify factors affecting lymph node metastasis (N1) in CCA. All statistical analyses were performed using STATA 10.1 software (StataCorp LLC, TX, USA). Statistical significance is defined as a $P$-value $< 0.05$.

## Results

### Evaluation of XB130 expression levels in CCA cell lines and an immortal cholangiocyte cell line

XB130 expression levels in the immortal cholangiocyte (MMNK1) and four CCA cell lines (KKU-100, KKU-213C, KKU-213A, and KKU-023) were measured semi-quantitatively using Western blot analysis. The results showed that XB130 was not expressed in MMNK1 cells (Fig 1). Among four studied CCA cell lines, XB130 was highly expressed in the KKU-213A cell line, but it was not expressed in other cells lines (*i.e.*, KKU-100, KKU-213C and KKU-023 cell lines).

### Roles of XB130 in proliferation, migration and invasion in CCA

The effects of down-regulation of XB130 by siRNA on cell proliferation, migration and invasion were examined in KKU-213A cell line. *XB130* gene silencing using siRNA technology successfully suppressed the expression levels of XB130 compared with the scrambled control at 48 h after treatment (Fig 2A). The cell proliferation rates and cell invasion and migration activities of siXB130-treated KKU-213A cell line were significantly reduced compared to control cells (Fig 2B–2F).

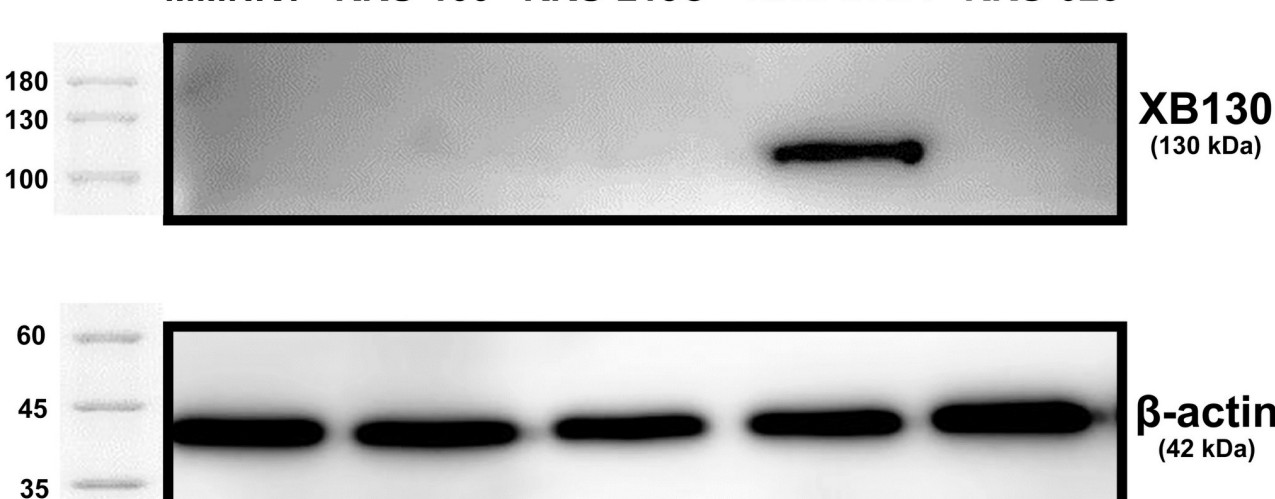

**Fig 1. Western blot analysis of XB130 in MMNK1, KKU-100, KKU-213C, KKU-213A and KKU-023 cell lines, with β-actin as a loading control.**

## XB130 activates the PI3K/Akt pathway and alters metastasis-associated proteins in CCA

To explore the mechanisms underlying XB130 expression, the possible downstream targets of XB130 including PI3K downstream (Akt), Ser473 phosphorylated Akt (pAkt) and epithelial-mesenchymal transition (EMT) markers (E-cadherin and vimentin) were measured in the XB130 knockdown-CCA cell lines using western blot analysis. The results indicated that XB130 siRNA effectively reduced XB130 protein levels; a phenomenon associated with decreased the expression and phosphorylation of Akt as well as the expression of EMT inducing protein (vimentin) as shown in Fig 2A. Our findings suggest that XB130 has an oncogenic property that can induce cell proliferation, migration and invasion in CCA via the activation of PI3K/Akt pathway leading to increasing of EMT processes.

## Expression levels of XB130 in CCA tissues

The expression levels of XB130 protein in CCA tissues (n = 151) were determined using immunohistochemistry. The protein was highly localized in cytoplasm of CCA cells, whereas it was weakly stained in normal bile duct and hepatocyte cells. The representative results of normal bile duct, hepatocyte and CCA tissues with XB130 expression levels were shown in Fig 3.

## Relationship between the XB130 expression levels and clinicopathological data of CCA patients

The relationship between XB130 protein expression levels and the clinicopathological features of CCA patients were summarized in Table 1. No relationship was observed between XB130 expression and the following variables: age ($\beta$ = -0.0023, $P$ = 0.976), gender ($\beta$ = 0.0042, $P$ = 0.958), histological type ($\beta$ = 0.0262, $P$ = 0.749), margin status ($\beta$ = -0.1305, $P$ = 0.110), moderate differentiation ($\beta$ = 0.1238, $P$ = 0.128), poor differentiation ($\beta$ = 0.1477, $P$ = 0.070) or M classification ($\beta$ = 0.074, $P$ = 0.365). The XB130 expression was significantly positive

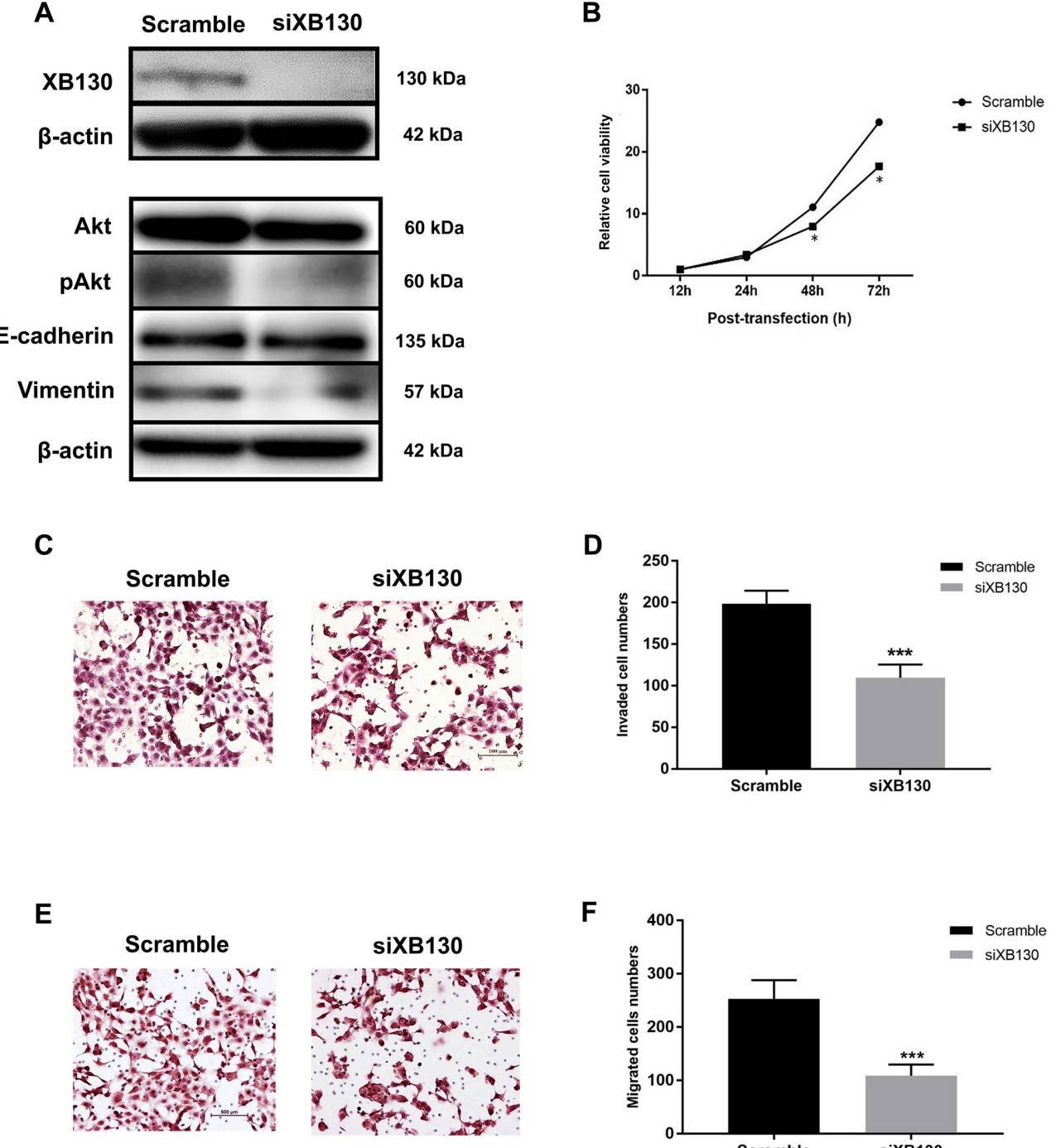

**Fig 2. Effects of siXB130 on proliferation, motility and invasiveness of KKU-213A cell line.** A): The expressions of XB130 (130 kDa), E-cadherin (~135 kDa), vimentin (~54 kDa), Akt (~60 kDa), pAkt (~60 kDa) in siXB130 and scramble groups measured by western blotting, with β-actin as a loading control. B): MTT assays of the scramble and siXB130 treated cells. C): Hematoxylin-staining invasive cells from cell invasion assays of the scramble and siXB130 treated cells. D): Graphical represents numbers of invasive cells from cell invasion assays. E): Hematoxylin staining migrated cells from cell migration assays. F): Graphical represents numbers of migrating cells from cell migration assays. The asterisk (*) indicates statistical significance at $P<0.05$ and asterisks (***) indicates statistical significance at $P < 0.001$.

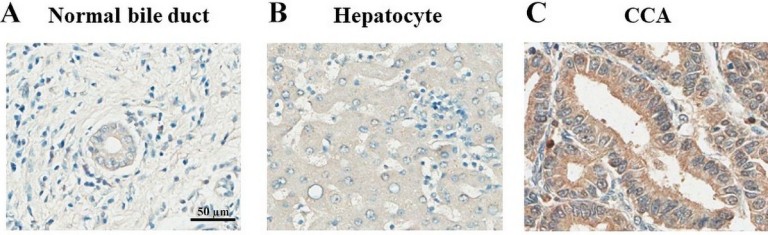

**Fig 3. XB130 expression patterns in: A) normal bile duct; intensity = 48, B) hepatocyte; intensity = 72 and C) CCA intensity = 220 analyzed by IHC.** The scale bar equals 50 μm.

**Table 1. Relationship between XB130 expression and clinicopathologic features of CCA patients.**

| Variable | XB130 expression | | | |
|---|---|---|---|---|
| | *Coef.* | *95% CI* | *β* | *P-value* |
| **Gender** | | | | |
| Female/male | -0.12 | -8.9111–8.6517 | 0.0042 | 0.976 |
| **Age** | | | | |
| Years | 0.41 | -15.4027–16.2285 | -0.0023 | 0.958 |
| **LVSI** | | | | |
| Yes/No | 22.08 | 6.7806–37.3836 | 0.2274 | 0.005 |
| **Histological type** | | | | |
| Tubular/Papillary type | 2.48 | -12.8509–17.8225 | 0.0262 | 0.749 |
| **Margin status** | | | | |
| Yes/No | -12.75 | -28.4395–2.9211 | -0.1305 | 0.110 |
| **Type of CCA** | | | | |
| Intrahepatic/Extrahepatic | 17.03 | 0.2762–33.8017 | 0.1623 | 0.046 |
| **Histologic Differentiation** | | | | |
| Well | | 1 | | |
| Moderated | 15.78 | -4.5935–36.1560 | 0.1238 | 0.128 |
| Poor | 50.16 | -4.1080–104.4313 | 0.1477 | 0.070 |
| **TNM staging** | | | | |
| I | | 1 | | |
| II | 10.58 | -10.0906–31.2698 | 0.0970 | 0.313 |
| III | 41.17 | 21.5696–60.7782 | 0.4048 | < 0.001 |
| IV | 28.41 | 6.2269–50.6001 | 0.2363 | 0.012 |
| **T classification** | | | | |
| T1 | | 1 | | |
| T2 | 3.93 | -11.8204–19.6827 | 0.0412 | 0.623 |
| T3 | 56.87 | 35.8058–77.9391 | 0.4314 | < 0.001 |
| T4 | 54.32 | 29.1228–79.5356 | 0.3327 | < 0.001 |
| **N classification** | | | | |
| N1/N0 | 19.71 | 4.5583–34.8619 | 0.2060 | 0.011 |
| **M classification** | | | | |
| M1/M0 | 8.91 | -10.494–28.3251 | 0.074 | 0.365 |

*Definition of TNM staging system described in the S1 Data.

*Coef, coefficients; 95% CI, 95%confidence interval; β, correlation coefficient.

relationship with LVSI ($\beta$ = 0.2274, $P$ = 0.005), intrahepatic type of CCA ($\beta$ = 0.1623, $P$ = 0.046), stage III of TNM staging ($\beta$ = 0.4048, $P$ < 0.001), stage IV of TNM staging ($\beta$ = 0.2363, $P$ = 0.012), T3 classification ($\beta$ = 0.4314, $P$ < 0.001), T4 classification ($\beta$ = 0.3327, $P$ < 0.001) and lymph node metastasis ($\beta$ = 0.2060, $P$ = 0.011).

## Univariate and multivariate analyses of clinicopathological variables influencing survival and lymph node metastasis in CCA

Clinicopathological parameters of 151 CCA patients (95 males and 56 females) with an age range of 34–79 years were analyzed. The impact of 12 variables, including age, gender, type of CCA, histological type, histologic differentiation, LVSI, T classification, N classification (lymph node metastasis), M classification, TNM staging, surgical margins and XB130 expression, on survival was investigated using univariate and multivariate analyses. On univariate analysis using the Cox's proportional hazards model, the poor survival was associated with the following parameters: LVSI (HR = 1.6150, $P$ = 0.007), tubular histological type (HR = 1.6348, $P$ = 0.005), positive margin status (HR = 1.6777, $P$ = 0.003), TNM staging (stage IV, HR = 4.2664, $P \leq$ 0.001), T classification (T4, HR = 2.0254, $P$ = 0.028), N1 classification (HR = 2.3754, $P \leq$ 0.001) and M1 classification (HR = 3.4339, $P \leq$ 0.001), but not with gender (HR = 1.0722, $P$ = 0.659), age (HR = 1.0139, $P$ = 0.163), intrahepatic type of CCA (HR = 1.1532, $P$ = 0.451), moderate differentiation, (HR = 1.5331, $P$ = 0.055), poor differentiation, (HR = 0.4662, $P$ = 0.287), T classification (T2, HR = 1.3788, $P$ = 0.111), T classification (T3, HR = 1.1805, $P$ = 0.543), TNM staging (stage II, HR = 1.2338, $P$ = 0.400), TNM staging (stage III, HR = 1.4768, $P$ = 0.101), XB130 expression (HR = 1.000, $P$ = 0.356) (Table 2). The Kaplan-Meier survival curves showed the comparison of the survival rate in 151 cases of CCA patients between gender, histological type, type of CCA, M classification, LVSI, margin status, histologic differentiation, TNM staging, T classification and N classification. Tubular histological type ($P$ = 0.0052), M1 classification ($P$ < 0.0001), LVSI ($P$ = 0.0062), positive margin status ($P$ = 0.0041), stage IV of TNM staging ($P$ < 0.001) and N1 classification ($P$ < 0.001) were significant related to prognosis as shown in S8 Fig.

In addition, multivariate analysis using Cox's proportional hazards model showed that only N1 classification (HR = 1.8707, $P$ = 0.002) and M1 classification (HR = 2.3925, $P$ < 0.001) were significantly correlated with poor survival. The two clinicopathologic variables, *i.e.*, N and M classifications, can be used as independent prognostic factors (Table 3). Moreover, the results obtained from multivariate analysis using logistic regression demonstrated that M1 and LVSI were independent factors for lymph node metastasis (Table 4).

## Discussion

The main finding of this study was that XB130 overexpression in CCA was related to cell proliferation, migration, and invasion, leading to CCA progression with aggressive clinical outcomes. These results suggest that the expression levels of XB130 in CCA cell lines, XB130 was highly expressed in the KKU-213A cell line, but it was not expressed in other cells lines (*i.e.*, KKU-100, KKU-213C and KKU-023 cell lines). The expression levels of XB130 in CCA cell lines may not be involved in the transformation from cholangiocytes to CCA cells. According to the cell line's properties, KKU-213A has the highest proliferation, invasion and migration rates among the other tested cell lines [22, 30]. Thus, in CCA, XB130 may play a critical role in tumor cell progression, but not in tumorigenesis. The KKU-213A cell line was selected for further functional analysis of XB130 using specific siRNA technique. Significant inhibition of cell proliferation, migration, and invasion caused by *XB130* silencing in KKU-213A cells suggests

**Table 2. Univariate analysis (Cox's proportional hazards model) for overall survival of CCA patients.**

| Characteristics | Univariate | | |
|---|---|---|---|
| | **HR** | **95% CI** | **P-value** |
| **Gender** | | | |
| Female/male | 1.0722 | 0.7569–1.5187 | 0.659 |
| **Age** | | | |
| Years | 1.0139 | 0.9943–1.0339 | 0.163 |
| **LVSI** | | | |
| Yes/No | 1.6150 | 1.1404–2.2870 | 0.007 |
| **Histological type** | | | |
| Tubular /Papillary type | 1.6348 | 1.1583–2.3073 | 0.005 |
| **XB130 expression** | | | |
| Intensity | 1.0000 | 0.9980–1.0053 | 0.356 |
| **Margin status** | | | |
| Yes/No | 1.6777 | 1.1873–2.3708 | 0.003 |
| **Type of CCA** | | | |
| Intrahepatic/Extrahepatic | 1.1534 | 0.7924–1.6788 | 0.451 |
| **Histologic Differentiation** | | | |
| Well | | 1 | |
| Moderated | 1.5331 | 0.9907–2.3722 | 0.055 |
| Poor | 0.4662 | 0.1144–1.8988 | 0.287 |
| **TNM staging** | | | |
| I | | 1 | |
| II | 1.2338 | 0.7559–2.0137 | 0.400 |
| III | 1.4768 | 0.9266–2.3538 | 0.101 |
| IV | 4.2664 | 2.4784–7.3443 | < 0.001 |
| **T classification** | | | |
| T1 | | 1 | |
| T2 | 1.3788 | 0.9291–2.0461 | 0.111 |
| T3 | 1.1805 | 0.6919–2.0140 | 0.543 |
| T4 | 2.0254 | 1.0781–3.8048 | 0.028 |
| **N classification** | | | |
| N1/N0 | 2.3754 | 1.6709–3.3769 | < 0.001 |
| **M classification** | | | |
| M1/M0 | 3.43396 | 2.1942–5.3741 | < 0.001 |

that XB130 plays an important role in the progression of CCA by promoting cell growth, migration, and invasion.

These findings are consistent with previous studies on XB130 in a variety of cancers, *e.g.*, thyroid tumor [19, 31], prostate cancer [15], lung cancer [17], gastric cancer [32], breast cancer

**Table 3. Multivariate analysis (Cox's proportional hazards model) for overall survival of CCA patients.**

| Characteristics | Multivariate | | |
|---|---|---|---|
| | **HR** | **95% CI** | **P-value** |
| **N classification** | | | |
| N1/N0 | 1.8707 | 1.2589–2.7797 | 0.002 |
| **M classification** | | | |
| M1/M0 | 2.3925 | 1.4589–3.9237 | 0.001 |

Table 4. Univariate and multivariate analyses (logistic regression model) of factor correlated with lymph node metastasis (N1).

| Clinicopathologic characteristics | Univariate | | | Multivariate | | |
|---|---|---|---|---|---|---|
| | Odds ratio | 95% CI | *P*-value | Odds ratio | 95% CI | *P*-value |
| **M category (+)** | 29.84 | 6.7488–131.9561 | < 0.001 | 22.17 | 4.9082–100.1789 | < 0.001 |
| **LVSI (+)** | 5.32 | 2.4970–11.3455 | < 0.001 | 3.69 | 1.6187–8.4202 | 0.002 |

[33], colorectal cancer [34], hepatocellular carcinoma [35], and skin tumor [24]. XB130 has been suggested to play significant roles in cancer progression through PI3K/Akt signaling pathway, leading to an increase in tumorigenic properties, including cell growth, migration, invasion and epithelial–mesenchymal transition (EMT) process [17, 31, 35–37]. It is worth noting that in CCA, the PI3K/Akt signaling pathway was constitutively activated and was involved in cell proliferation, migration and invasion [38, 39]. Suppression of the PI3K/Akt signaling pathway by various anti-cancer drugs could also inhibit CCA cell progression [35, 40–43]. From the above mentioned, we hypothesized that XB130 activates PI3K/Akt pathway resulting in the phosphorylation of Akt (pAkt) and induction of EMT process in CCA cells. To prove the hypothesis, the detections of PI3K/Akt pathway-downstream marker (pAkt) and EMT markers (E-cadherin and vimentin) expression levels in siXB130 and the vertical control cells were measured using western blot analysis. The results showed that pAkt and vimentin expression levels were decrease in the XB130 knockdown-CCA cell line. Therefore, we concluded that the overexpression of XB130 in CCA accelerates the cancer progression via PI3K/Akt signaling pathway.

The XB130 expression levels in clinical tissues were further analyzed to assess its potential for being used as an accurate prediction of survival of patients with CCA and as a valid biological indicator for the aggressiveness of CCA. The XB130 expression level was not an independent prognosis factor for survival in CCA. It should be noted that distant metastasis (M1-classification) has more potential clinical use to predict for the risk of lymph node metastasis in CCA than the LVSI. The XB130-mediated aggressiveness of CCA was confirmed by the biological function assays, *i.e.*, cell proliferation, invasion, and migration in cell lines. In CCA tissues, the elevated XB130 expression levels had positive relationship with lymphovascular space invasion (LVSI), intrahepatic type of CCA, high TNM staging (stage III, IV), high T classification (T3, T4), and N1 classification (lymph node metastasis). The high T classification, LVSI, lymph node metastasis, and high TNM staging all indicate tumor extension and invasion, which supported the *in vitro* investigation of cell proliferation and invasion in KKU-213A cells. The Kaplan-Meier survival curves showing LVSI, margin status, TNM staging, and N classification were significantly related to prognosis. Although, XB130 level was not directly associated with survival outcome in CCA patients, but it was associated with other factors correlated with survival, including LVSI, high TNM staging and lymph node metastasis. Therefore, our findings demonstrated that the adapter protein XB130 is a significant player in CCA progression and metastasis.

## Conclusion

In this work, the expression of XB130 was assessed along with its impact on CCA proliferation, migration, and invasion *in vitro*. Furthermore, high expression levels of XB130 in tissues reflect a role in predicting the aggressive behavior of CCA, as evidenced by significant risk factors, i.e., LVSI, lymph node metastasis, and high TNM staging in the patients. Targeting XB130 may assist in slowing tumor growth and decreasing cell migration and invasion. Therefore, high XB130 expression might be considered as an indicator of aggressive CCA and a potential target for CCA therapy.

## Supporting information

**S1 Fig. Original image of western blot result of XB130 expression in cell lines.**
(PDF)

**S2 Fig. Original image of western blot result of β-actin expression in cell lines.**
(PDF)

**S3 Fig. Original image of western blot result of phosphorylation of Akt expression in media, scramble and siXB130 cells.**
(PDF)

**S4 Fig. Original image of western blot result of Akt expression in media, scramble and siXB130 cells.**
(PDF)

**S5 Fig. Original image of western blot result of E-cadherin expression in media, scramble and siXB130 cells.**
(PDF)

**S6 Fig. Original image of western blot result of vimentin expression in media, scramble and siXB130 cells.**
(PDF)

**S7 Fig. Original image of western blot result of β-actin in media, scramble and siXB130 cells.**
(PDF)

**S8 Fig. The Kaplan-Meier analysis of clinical data.**
(PDF)

**S1 Data. Definition of TNM staging system.**
(PDF)

**S2 Data. Evaluation of immunohistochemical staining by ImageJ.**
(PDF)

**S1 Appendix.**
(PDF)

## Acknowledgments

We would like to thank Srinagarind Hospital for providing the specimens. Cholangiocarcinoma Research Institute (CARI) was also acknowledged for providing various cell lines for conducting experiments. We would like to acknowledge Prof. Yukifumi Nawa for editing the manuscript via Publication Clinic KKU, Thailand. We thank Prof. Reza Alaghehbandan for editing the manuscript. We thank Prof. Somchai Pinlaor for providing the antibodies that were used in this work. We thank Dr. Uthumporn Domthong and Mrs. Kaewjai Tepsuthumarat for their statistical advice and guidance.

## Author Contributions

**Conceptualization:** Raynoo Thanan, Piti Ungarreevittaya.

**Data curation:** Pirawan Poosekeaw, Piti Ungarreevittaya.

**Funding acquisition:** Raynoo Thanan, Piti Ungarreevittaya.

**Investigation:** Pirawan Poosekeaw.

**Methodology:** Sitthichai Iamsaard, Raynoo Thanan.

**Resources:** Chawalit Pairojkul, Banchob Sripa, Prakasit Sa Ngiamwibool, Sitthichai Iamsaard, Chadamas Sakonsinsiri, Raynoo Thanan.

**Supervision:** Piti Ungarreevittaya.

**Writing – original draft:** Pirawan Poosekeaw.

**Writing – review & editing:** Chawalit Pairojkul, Banchob Sripa, Prakasit Sa Ngiamwibool, Sitthichai Iamsaard, Chadamas Sakonsinsiri, Raynoo Thanan, Piti Ungarreevittaya.

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
