## [Decision Letter · Decision Letter 0]

2 Feb 2021

PONE-D-20-37191

Adaptor protein XB130 regulates the aggressiveness of cholangiocarcinoma

PLOS ONE

Dear Dr. Ungareevittaya,

Thank you for submitting your manuscript to PLOS ONE. After careful consideration, we feel that it has merit but does not fully meet PLOS ONE’s publication criteria as it currently stands. Therefore, we invite you to submit a revised version of the manuscript that addresses the points raised during the review process.

At the end of this letter you will find the reports and comments. Please read also carefully the suggested literature.and remind that you are free to choose , which references you will cite or not, if at all.    

We look forward to receiving your revised manuscript.

Kind regards,

Konradin Metze

Academic Editor

PLOS ONE

Additional Editor Comments :

The investigation is original and quite interesting but there are important points to be considerably improved before the paper can be considered for publication:

The authors categorized a lot of variables initially expressed as cardinal numbers (which causes an important loss of information and may induce considerable bias).In particular,  the immunohistochemical quantification should also be expressed in cardinal numbers and not dichotomized. .

We are very aware, that this kind of “data treatment” is still popular among pathologists, but it is a great source of error and publications with low credibility. Therefore, all calculations must be repeated with cardinal numbers. The results might be different.

In a final step, the authors may suggest a score after categorizing of one or another variable, but this must be clearly justified both from a mathematical and a biological point of view and a comparison of the calculi done with and without categorizing must be given. (please see the last paper  for how to do this).

Please consult the following literature:

Dichotomizing continuous predictors in multiple regression: a bad idea. Stat Med 25: 127-141, 2006

Pitfalls in the assessment of prognostic factors. Lancet Oncol. 2011 Nov;12(12):1095-6.

Dichotomizing continuous prognostic factors can cause paradoxical results in survival models. J Am Coll Surg. 2011 Jan;212(1):132-4.

Dichotomization of continuous data--a pitfall in prognostic factor studies. Pathol Res Pract. 2008;204(3):213-4

Breaking up is hard to do: the heartbreak of dichotomizing continuous data. Can J Psychiatry 2002;47:262– 66.

A simple score derived from bone marrow immunophenotyping is important for prognostic evaluation in myelodysplastic syndromes  Sci Rep. 2020 Nov 20;10(1):20281.

Konradin Metze, Academic Editor

Journal Requirements:

2. As part of your revisions, please indicate in your Methods statement whether you obtained any other type of authorization, consent or approval for the use of the leftover tissues following pathological diagnosis of CCA. This may include patient consent, consent from next of kin, and so forth. Thank you for your cooperation in this matter.

Reviewers' comments:

Reviewer's Responses to Questions

**Comments to the Author**

1. Is the manuscript technically sound, and do the data support the conclusions?

Reviewer #1: Partly

Reviewer #2: Yes

Reviewer #3: Partly

2. Has the statistical analysis been performed appropriately and rigorously? 

Reviewer #1: Yes

Reviewer #2: Yes

Reviewer #3: Yes

3. Have the authors made all data underlying the findings in their manuscript fully available?

Reviewer #1: Yes

Reviewer #2: Yes

Reviewer #3: Yes

4. Is the manuscript presented in an intelligible fashion and written in standard English?

Reviewer #1: Yes

Reviewer #2: Yes

Reviewer #3: Yes

5. Review Comments to the Author

Reviewer #1: The artical is clearly written with minimal spelling and grammatical error.

Very little data regarding mechanism of action is presented. The authors show that a CCA cell line with high XB130 expression has faster proliferation, more motility and more extensive invasiveness than a normal or two CCA cell lines with low XB130 expression using in vitro assays. The authors also show that these properties can be suppressed in the high expresser by siRNA knockdown of XB130 as compared to a scrambled control. What is not presented is why XB130 expression promotes these activities. Several pathways downstream of XB130 are mentioned in the introduction. The authors should inhibit individual downstream pathways (e.g., PI3K/Akt) to test whether such inhibition recapitulates the effect of siRNA knockdown of XB130. The authors should also over-express XB130 in MMNK1 cells and XB130-low CCA cells to test whether this expression is sufficient to confer the proliferative, motility and invasiveness characteristics of XB130-high CCA cells. There is published data indicating that XB130 can promote tumorigenesis via activation of the PI3K/Akt pathway, but it would improve the manuscript to show that in this system. As it is now, too much of the manuscript's mechanistic conclusions rest on a single CCA cell line. More controls and more mechanistic studies are needed for this aspect.

For use as a biomarker, the authors need to report specificity and sensitivity. Discussion should also be provided that compares the sensitivity and specificity of XB130 against existing CCA biomarkers of which there are now a great number.

Table 1 should include definitions of staging features:

N0 (No regional lymph node metastasis)

N1 (Regional lymph node metastasis)

M0 (No distant metastasis)

M1 (Distant metastasis)

T1 (Solitary tumor; no vascular invasion)

T2a (Solitary tumor with vascular invasion)

T2b (Multiple tumors)

T3 (Tumor that has perforated the visceral peritoneum or otherwise demonstrating invasivity)

T4 (Tumor with periductal invasion).

Stage I: T1+N0+M0

Stage II: T2+N0+M0

Stage III: T3+N0+M0

Stage IVA: T4+N0+M0

Stage IVB: T(1-4)+N1 or T(1-4)+M1

Reviewer #2: The authors analyze the expression of the adaptor protein XP130 in 151 cases with gall bladder carcinoma. They note poor survival rates in cases with high XP 130 expression, which they confirm by studies of 4 cell lines. This is a good article and should be published provided that

1. the discrepancies in table 1 (M0, M1<> TNM); (N0, N1<> Lymph node metastases) etc. should be explained 7 corrected

2. What about pTNM?, Tumor size, organ of metastasis (liver, others?)

3. The collection time (2007 - 2017) is quite long; what about changes of the TNM classification during that time?

4. The aspect of tumor cell heterogeity should be briefly adressed (for example see and cite: VULETIĆ, Filip et al. Intratumoral heterogeneity. Diagnostic Pathology, [S.l.], v. 4, n. 1, feb. 2018. ISSN 2364-4893. Available at: <http: 257="" article="" content="" dpath="" index.php="" view="" www.diagnosticpathology.eu="">5. The IHC color classification threshold / method should be explained more in detail (which color space, which color adjustment / image quality analysis has been performed (for details see and cite: KAYSER, Klaus et al. Texture and object related image analysis in microscopic images. Diagnostic Pathology, [S.l.], may 2015. ISSN 2364-4893. Available at: <http: 14="" article="" content="" dpath="" index.php="" view="" www.diagnosticpathology.eu="">.</http:></http:>

Reviewer #3: To the editor/authors:

The study by Poosekeaw et al, deals with the role of the adaptor protein XB130 in cholangiocarcinoma. The authors present data using 5 different cell lines, and one of them expressing XB130. For further experiments, they have used XB130-expressing cell line and via siRNA they knocked down XB130. They could show that XB130 plays a role in proliferation, migration and invasion, at least in this cell line. Additional, they have analysed XB130 expression in 151 CCA tissue samples and could find a correlation of XB130 expression, M classification and LVSI.

Although of interest, following improvements have to be done before it can be considered for publication in Plos One.

Critical points:

1) Figure 1: First, the authors should mention in the fig-legend that tubulin serves as a loading control. Second, please indicate kDa of target protein in brackets. Furthermore, it would also be interesting to know why the other cell do not express XB130 at all. Have the authors checked mRNA expression? Is there a post-transcriptional modification? Are there any mutations or epigenetically silencing in/of XB130 known?

2) Figure 2: General for Fig2A-F. It would be a nice control to see all parameters in untreated KKU-213A (meaning without any transfection). Next point: the authors show a complete down regulation after 48h but, if I understood the data right, the results of proliferation, migration and invasion were analysed after additional 24h? In order to check the efficiency of the XB130 knock down a time course would be helpful – Fig2A: Western Blot after 24, 48 and 72h post-transfection. An additional control experiment would be to use a XB130 non-expressing cell line (e.g. KKU213C).

3) Figure 3: Please give the scale bar! Furthermore, it would be helpful to mark the CCA cells (e.g. with a *), normal bile duct (e.g. with an arrow) and hepatocytes (e.g. with an arrowhead) for non-histologists.

4) Fig 4, legend: please indicate the number of patients (n)

5) Have the authors checked for downstream targets, like PI3K or Akt? These experiments could be easely done by Western Blotting and would underline the specific role of XB130 in tested cell lines and in FFPE material (in this case testing by IHC).

6) A further suggestion/question: are data available for Ki67 level in the FFPE samples?

7) Have the authors checked, both cell lines (WB) and FFPE (IHC), EMT-related targets (N-, E-cadherin, vimentin, or EMT-TFs, e.g. TWIST, SNAI, ZEB)? I think these experiments would improve the paper and underline the conclusions the authors have done!

6. PLOS authors have the option to publish the peer review history of their article (what does this mean?). If published, this will include your full peer review and any attached files.

Reviewer #1: No

Reviewer #2: No

Reviewer #3: No

---

## [Author Response · Author response to Decision Letter 0]

29 Mar 2021

Please look at the response to reviewers letter in attached file

---

## [Decision Letter · Decision Letter 1]

19 Apr 2021

PONE-D-20-37191R1

Adaptor protein XB130 regulates the aggressiveness of cholangiocarcinoma

PLOS ONE

Dear Dr. Ungareevittaya,

Thank you for submitting your manuscript to PLOS ONE. After careful consideration, we feel that it has merit but does not fully meet PLOS ONE’s publication criteria as it currently stands. Therefore, we invite you to submit a revised version of the manuscript that addresses the points raised during the review process.

Please read carefully my ccomments below !

We look forward to receiving your revised manuscript.

Kind regards,

Konradin Metze

Academic Editor

PLOS ONE

Academic Editor´s comments :

I am not satisfied with the changes. There are still variables, measured originally in cardinal numbera, but tater categorized , such as age. Why  is there a cut point at 50 years, instead of 45, 47 , 55 etc. ?

 This decision is completely arbitraly and therefor unscientific.

Before applying  quartiles, the authors should calculate with the cardinal numbers of raw data  This applies to the correlations, as well as to Cox and logistic regressions !

The  graphic  in the reply to the editor should be i n t e g r a  t e d  into the manuscript.  Regarding this graphhic  please try non-linear curve-fittings , perhaps a logarithmic  or , perhaps still better, a hyperbolic  function  and give the Rsquares as estimates of goodness-of –fit !.  Please discuss this result ! 

In summary : You should calculate all models first with continus cardinal numbers whenever possible.  In a

s e c o n d  step you can establish models  using the quertiles, but always with a l l  other variables measured

in cardinal numbers  n o f categorized.

Please sen a copy m where you highlight your  changes.

Konradin Metze, Academic Editor 

Reviewers' comments:

Reviewer's Responses to Questions

**Comments to the Author**

1. If the authors have adequately addressed your comments raised in a previous round of review and you feel that this manuscript is now acceptable for publication, you may indicate that here to bypass the “Comments to the Author” section, enter your conflict of interest statement in the “Confidential to Editor” section, and submit your "Accept" recommendation.

Reviewer #1: All comments have been addressed

Reviewer #2: (No Response)

Reviewer #3: All comments have been addressed

2. Is the manuscript technically sound, and do the data support the conclusions?

Reviewer #1: Yes

Reviewer #2: (No Response)

Reviewer #3: Yes

3. Has the statistical analysis been performed appropriately and rigorously? 

Reviewer #1: Yes

Reviewer #2: (No Response)

Reviewer #3: Yes

4. Have the authors made all data underlying the findings in their manuscript fully available?

Reviewer #1: Yes

Reviewer #2: (No Response)

Reviewer #3: Yes

5. Is the manuscript presented in an intelligible fashion and written in standard English?

Reviewer #1: Yes

Reviewer #2: (No Response)

Reviewer #3: Yes

6. Review Comments to the Author

Reviewer #1: (No Response)

Reviewer #2: The authors have responded to all recommendations of the reviewers, and the article should be published in its revised form.

Reviewer #3: The authors have answered all my questions and comments. Furthermore, the could nicely show the downstream effects of XB130 in the cell line. To my hand the manuscript can be accepted for publishing in PlosOne.

7. PLOS authors have the option to publish the peer review history of their article (what does this mean?). If published, this will include your full peer review and any attached files.

Reviewer #1: No

Reviewer #2: No

Reviewer #3: No

---

## [Author Response · Author response to Decision Letter 1]

30 May 2021

Please see the cover letter and the response to Editor letter.

---

## [Decision Letter · Decision Letter 2]

5 Jul 2021

PONE-D-20-37191R2

Adaptor protein XB130 regulates the aggressiveness of cholangiocarcinoma

PLOS ONE

Dear Dr. Ungareevittaya,

Thank you for submitting your manuscript to PLOS ONE. After careful consideration, we feel that it has merit but does not fully meet PLOS ONE’s publication criteria as it currently stands. Therefore, we invite you to submit a revised version of the manuscript that addresses the points raised during the review process.

Please corect according to the suggestions you find below !

We look forward to receiving your revised manuscript.

Kind regards,

Konradin Metze

Academic Editor

PLOS ONE

Additional Editor Comments 

**The paper has been improved. Problems are still found in  table 1 . Are these data really all correlations ?Which kind of correlation ? Spearman ? Kendall ? Pearson ? The correlation coefficients are all missing  ! Please add the coefficients together with their confiidence intervals in all cases with p<0.05**

**Konradin Metze, Academic Editor **

Reviewers' comments:

Reviewer's Responses to Questions

**Comments to the Author**

1. If the authors have adequately addressed your comments raised in a previous round of review and you feel that this manuscript is now acceptable for publication, you may indicate that here to bypass the “Comments to the Author” section, enter your conflict of interest statement in the “Confidential to Editor” section, and submit your "Accept" recommendation.

Reviewer #1: All comments have been addressed

Reviewer #2: All comments have been addressed

Reviewer #3: All comments have been addressed

2. Is the manuscript technically sound, and do the data support the conclusions?

Reviewer #1: Yes

Reviewer #2: Yes

Reviewer #3: Yes

3. Has the statistical analysis been performed appropriately and rigorously? 

Reviewer #1: Yes

Reviewer #2: Yes

Reviewer #3: Yes

4. Have the authors made all data underlying the findings in their manuscript fully available?

Reviewer #1: Yes

Reviewer #2: Yes

Reviewer #3: Yes

5. Is the manuscript presented in an intelligible fashion and written in standard English?

Reviewer #1: Yes

Reviewer #2: Yes

Reviewer #3: Yes

6. Review Comments to the Author

Reviewer #1: All of my past concerns were addressed in the prior review.

The current submission addresses questions regarding statistical methodology made by a different reviewer. I am satisfied with the revisions to the statistics provided in the current submission.

All data was and is provided. The new submission additionally has the blots used to generate the figures of the manuscript.

The English is intelligible and written in standard English

Reviewer #2: no comments can be made, a system's error is very likely. Unfortunately, I cannot be of any additional assistance.

Reviewer #3: (No Response)

7. PLOS authors have the option to publish the peer review history of their article (what does this mean?). If published, this will include your full peer review and any attached files.

Reviewer #1: No

Reviewer #2: No

Reviewer #3: No

---

## [Author Response · Author response to Decision Letter 2]

23 Aug 2021

Please see the response to editor file

---

## [Editor Report · Decision Letter 3]

13 Oct 2021

Adaptor protein XB130 regulates the aggressiveness of cholangiocarcinoma

PONE-D-20-37191R3

Dear Dr. Ungareevittaya,

We’re pleased to inform you that your manuscript has been judged scientifically suitable for publication and will be formally accepted for publication once it meets all outstanding technical requirements.

Kind regards,

Konradin Metze

Academic Editor

PLOS ONE
---

## [Editor Report · Acceptance letter]

2 Nov 2021

PONE-D-20-37191R3 

Adaptor protein XB130 regulates the aggressiveness of cholangiocarcinoma 

Dear Dr. Ungarreevittaya:

I'm pleased to inform you that your manuscript has been deemed suitable for publication in PLOS ONE. Congratulations! Your manuscript is now with our production department. 

Kind regards, 

on behalf of

Prof. Konradin Metze 

Academic Editor

PLOS ONE